# How Does Pseudo-Jahn-Teller Effect Induce the Photoprotective Potential of Curcumin?

**DOI:** 10.3390/molecules28072946

**Published:** 2023-03-25

**Authors:** Dagmar Štellerová, Vladimír Lukeš, Martin Breza

**Affiliations:** Institute of Physical Chemistry and Chemical Physics, Slovak University of Technology in Bratislava, Radlinského 9, SK-812 37 Bratislava, Slovakia

**Keywords:** DFT, natural compounds, tautomers, molecular symmetry, electron transitions

## Abstract

In this paper, the molecular and electronic structure of curcumin is studied. High-symmetric gas-phase tautomers and their deprotonated forms in various symmetry groups are identified. The stability of lower-symmetry structures was explained by using the Pseudo-Jahn-Teller (PJT) effect. This effect leads to stable structures of different symmetries for the neutral enol and keto forms. The presented analysis demonstrated the potential significance of the PJT effect, which may modulate the setting of electronic and vibrational (vibronic) energy levels upon photodynamic processes. The PJT effect may rationalize the photoprotection action and activity of naturally occurring symmetric dyes.

## 1. Introduction

Symmetry represents a fundamental concept in physics, chemistry, and biology. Molecular symmetry explains various physical and chemical properties of compounds, and their toxicity. There are few types of biomolecules in plants and fungi that are symmetric. It is a well-known fact that high-symmetric xenobiotics may perform a negative role in living systems. For example, benzene, pyrene, or the next polycyclic hydrocarbons are highly carcinogenic [1]. If an aromatic entity is necessary in some biochemical processes, the symmetry is lowered by substituting side chains using, e.g., an alkyl, acetate, hydroxyl, alkoxy, vinyl, or acetate group. Cyclic compounds derived from symmetric tetrapyrrole porphin can be mentioned in this connection [2].

Curcumin (diferuloylmethane) has an interesting position among symmetric organic molecules in plants. This yellow pigment of turmeric (*Curcuma longa*) has been found to have antioxidant, antibacterial, anti-inflammatory, and antitumor activity in a variety of animal models of human diseases [3,4,5]. Curcumin has already entered clinical trials, which have demonstrated its high-dose nontoxicity and exhibited poor bioavailability in humans [6]. More recently, several studies have examined the protective effects of curcumin against skin carcinogenesis caused by chronic UV irradiation [7,8]. These are related to the diarylheptanoids structure, which consists of two aromatic rings (aryl groups) joined by a seven-carbon chain (heptane) and having various substituents. Its diketone tautomer (1E,6E)-1,7-bis(4-hydroxy-3-methoxyphenyl)hepta-1,6-diene-3,5-dione exists in keto–keto forms (Figure 1a,b) in water and in enol–keto forms (Figure 1c,d) in organic solvents [9,10,11]. Enols are readily deprotonated to form enolates (Figure 1e,f). Due to its hydrophobic nature, curcumin is poorly soluble in water, slightly soluble in aliphatic or alicyclic organic solvents, but easily soluble in polar organic solvents [12]. Interestingly, the solid phase [13] contains only the enol–keto tautomeric form, which can form polymorphic crystals [14].

The effect of curcumin on acute photo-damage induced by higher dose UV irradiation is still unclear. In principle, its mechanism can be based on fluorescence, phosphorescence, and/or nonradiative transitions. In general, several parallel processes may be included in a photoprotectivity action. Upon the absorption of a high-energy dose, a low-energy photon may be generated because of the luminescence. Alternatively, the excited-state curcumin can interact with the highly reactive singlet oxygen molecule with the subsequent formation of the less reactive triplet oxygen molecule [15]. Such processes require the appropriate setting of energy levels of electronic and vibration (vibronic) states. From a theoretical point of view, the (Pseudo)-Jahn-Teller effect (PJTE) can particularly modulate these energy levels of symmetric compounds. Contrary to the classical Jahn-Teller effect (JTE) describing the vibronic interaction between degenerate electronic states, the PJTE is related to the vibronic interaction between two pseudo-degenerate electronic states. In some cases, the vibronic interaction within excited electronic states can be so strong that the energy surface of the lower state penetrates the ground-state adiabatic potential surface (APS) producing additional (global) minima in which the system is distorted. Such a situation is denoted as hidden JTE (for degenerate excited state) or hidden PJTE (for pseudo-degenerate excited states) [16]. The hidden JTE was first demonstrated in the structure and properties of the ozone molecule [17]. The electronic ground state of this molecule in a high-symmetry regular triangular geometry is non-degenerate. Its excited double-degenerate electronic E state produces three equivalent lower-energy APS minima, in which the configuration of the molecule is obtuse triangular. 

(P)JTE is responsible for a variety of phenomena in chemical reactivity, spectroscopy, stereochemistry, crystal chemistry, molecular, and solid-state physics, and materials science [18]. This effect has been demonstrated for a variety of metal complexes, inorganic, and organic compounds of different sizes [19,20,21,22,23]. In this context, we can mention the research of photosynthetic conversion of water to molecular oxygen in Photosystem II occurring in cyanobacteria, algae, and plants. The photosynthetic process is catalyzed by the inorganic Mn_4_CaO_5_ cluster, where the orientation of the Mn^III^ bonding is ascribed to the orientational pseudo-Jahn-Teller isomerism [24]. However, we have not found any research in the literature on this treatment of naturally occurring dyes to date.

With respect to the facts mentioned above, we decided to present the theoretical DFT (Density functional theory) [25,26] calculations of curcumin tautomers (see Figure 1). This molecule represents a suitable model of a symmetric natural dye molecule where the PJTE can appear. The partial aims of this study are systematic geometry optimization of the gas-phase structures in various symmetric groups and an explanation of their stable structures of lower symmetry using PJTE treatment. Finally, the theoretical results for optical spectra will be compared with the available experimental data. The possible photoprotective role of curcumin tautomers will be discussed.

## 2. Theory

According to the Jahn-Teller (JT) theorem [27], a non-linear configuration of atomic nuclei in a degenerate electron state is unstable. It implies the existence of at least one stable nuclear configuration of lower symmetry where the electron degeneracy is removed. An analytical formula for the APS of such systems can be obtained using a perturbation theory, and the atomic coordinates of the corresponding stable systems are evaluated by its minimization [28]. For large systems, this analytical formula for APS description is too complicated to be solved. In such a case, a group-theoretical treatment is used to predict the symmetry of the stable structures. Subsequently, geometry optimization within the predicted symmetry group is performed using any quantum-chemical method.

The method of step-by-step symmetry descent [29,30] is based on consecutive splitting of the degenerate electron state (i.e., its multidimensional irreducible representation) of a parent ‘unstable’ symmetry group simultaneously with the removal of symmetry elements. As the non-degenerate electron state (i.e., its one-dimensional irreducible representation) of an immediate subgroup is obtained, this is denoted as ‘stable’, and any further symmetry descent stops. Various symmetry descent paths are possible because no or only partial splitting might be obtained by symmetry descent to some immediate subgroups.

The epikernel principle method [31,32] represents an alternative treatment. It is based on the Jahn-Teller active distortion coordinate *Q* of Λ representation for a degenerate electron state *Ψ* of Γ representation within the parent symmetry group. Λ must belong to the non-totally symmetric part of the symmetrized direct product of Γ as implied by a non-vanishing value of $<\Psi \left|\frac{\partial^{}\hat{H}}{\partial Q}\right|\Psi >$ integrals, where $\hat{H}$ denotes Hamiltonian. Thus, the following relation must be valid:Λ ∈ [Γ ⊗ Γ] (1)

Totally symmetric vibrations (Λ = A, A_1_, A_1g_, or A’) do not change the symmetry of the system, and so the corresponding coordinates are not denoted as JT active. If the molecule of a symmetry group G undergoes the distortion *Q* of Λ representation, the molecule adopts the symmetry of its kernel or epikernel subgroup. The kernel subgroup K(G, Λ) of the parent group G for the *Q* coordinate of Λ representation is the minimal group that contains symmetry operations of G that leave the Λ representation invariant (i.e., with the same characters as the identity operation). The epikernel subgroup E(G, Λ) of the parent group G for the degenerate *Q* coordinate of the multidimensional Λ representation contains symmetry operations of the group G that leave the Λ representation invariant in a part of the distortion space (i.e., some components of this representation). In other words, epikernels are intermediate subgroups between the parent group and the kernel group. According to the epikernel principle, extremum points on a JT energy surface correspond to the kernels and epikernels of the parent group. As a rule, stable APS minima correspond to maximal epikernel symmetry. As this method is based on the perturbation theory with the linear terms of the Taylor expansion of the perturbation operator, $\left(\frac{\partial \hat{H}}{\partial Q}\right)∆Q$, it includes the most essential terms only and cannot predict the symmetries of all stable structures, e.g., groups containing C_5_ rotations [30].

Similar instability may be observed for pseudodegenerate electronic states. Sufficiently strong vibronic coupling between any two non-degenerate electronic states (often ground and excited) with an energy gap between them, which leads to instability and distortion of the polyatomic configuration, is known as the PJTE [22,28]. In the simplest case of two interacting electronic states *Ψ*_1_ and *Ψ*_2_ of different space symmetries, we obtain their APS in the form:(2)$$E\left(Q\right)=\frac{1}{2}KQ^{2}\pm [\frac{{∆}^{2}}{4}+F^{2}Q^{2}{]}^{1/2}$$
where *E* is the energy of the electronic state, *Q* is the distortion coordinate, Δ is the energy difference between the two electronic states in the undistorted geometry, *K* is the primary force constant (without vibronic coupling), and *F* is the vibronic coupling constant. If energy difference:(3)$$∆<2\frac{F^{2}}{K}$$
the curvature of the lower state is negative, which implies instability in the *Q* direction. Otherwise, *Q* = 0 for the stable structure and its symmetry is preserved despite the APS curvature being lowered (but positive). We cannot predict the maximal energy difference Δ between pseudodegenerate electronic states that cause the instability of PJT because it depends on the strength *F* of their mutual coupling (in relation to the *K* value), which depends on the system under study. The PJT interaction decreases with Δ and is restricted to the states of the same spin multiplicity only (because of the orthogonality of the spin states). In principle, PJT interactions (despite being very weak) are possible in any system of atomic nuclei, but PJTE is considered only in such systems where its consequences (in spectra, structure, reactivity, etc.) are observed. 

The epikernel principle method [31,32] can be used to predict the symmetry of the stable structures due to PJTE. In this case, the symmetry Λ of the JT active coordinate *Q* for non-vanishing values of $<\Psi_{1}\left|\frac{\partial^{}\hat{H}}{\partial Q}\right|\Psi_{2}>$ integrals are defined as: Λ ∈ Γ_1_ ⊗ Γ_2_(4)
where Γ_1_ and Γ_2_ are representations of electronic states *Ψ*_1_ and *Ψ*_2_, respectively, in the unperturbed parent geometry [33], see Appendix A.

## 3. Results

The highest identified possible structure of curcumin is that of C_2v_ symmetry group. We studied its symmetric neutral keto–keto (**I**) and enol–keto (**II** models) forms, and anionic enolate (III) forms. PJTE is connected with a symmetry decrease, so it cannot be observed in nonsymmetric molecules (C_1_ symmetry group). The PJT symmetry descent C_s_ → C_1_ or C_2_ → C_1_ is too simple and often can be considered as the final parts of more complex PJTE within their C_2v_ supergroup. Therefore, we investigate the C_2v_ symmetry containing the alkene linker in an all-trans configuration with both 4-hydroxy-3-methoxyphenolic groups either in anti- (**Ia**, **IIa**, and **IIIa** models) or in syn- (**Ib**, **IIb**, and **IIIb** models) conformations of methoxyl and central C-O groups. Geometry optimizations of these systems began in the C_2v_ symmetry group. Subsequently, their geometries were reoptimized within its C_2_, C_s_, and C_1_ subgroups. The obtained structures are briefly presented in Table 1 and visualized in Figure 1 and Appendix A. The electronic excited states for every optimized geometry were evaluated, and the first five lowest energy transitions are collected in Table 2. Equation (2) for PJT interaction is based on a perturbation principle, so we can suppose that the oscillator strengths *f* for the electron transitions between corresponding electron states in the parent group and its subgroups do not change significantly during PJT symmetry decrease. Thus, we use the *f* values to identify the corresponding excited states in the subgroups.

In the case of curcumin keto–keto forms **Ia** and **Ib**, we have found unstable geometries of C_2v_ symmetry with a single imaginary a_2_ vibration and stable C_2_ symmetry geometries without any imaginary vibration that correspond to their kernel subgroup (Appendix A, Appendix A). This symmetry descent can be explained by the PJT vibronic interaction between the ground electron state X^1^A_1_ and the second excited state 1^1^A_2_ within the C_2v_ symmetry. Due to symmetry reasons (see Appendix A), this electronic state corresponds to the second excited state 1^1^A within the C_2_ symmetry with an increased energy difference between both pseudo-JT interacting states, as proved by the corresponding excitation energy (see Table 2). This relation is confirmed by similar oscillator strengths corresponding to both excited states within structures of different symmetries. The **Ib** form is more stable and its pseudo-JT stabilization energy is higher as well (see Table 1).

Geometry optimization of curcumin enol forms **IIa** and **IIb** leads to unstable structures of symmetry C_2v_ with a single imaginary b_1_ vibration (Appendix A), which corresponds to the movement of central enolic hydrogen to oxygen, and stable geometries of C_s_ and C_1_ without imaginary vibrations (Table 1). The C_s_ symmetry group with the preserved σ_v_ mirror plane, which coincides with the molecular plane of curcumin and is denoted as C_s_(σ_v_), corresponds to the kernel group K(C_2v_, b_1_) (Appendix A). This symmetry descent can be ascribed to the PJT vibronic interaction between the ground electron state X^1^A_1_ and the first excited state 1^1^B_1_ within the C_2v_ symmetry (see Table 2 and Equation (3)). Due to symmetry reasons (see Appendix A), this electron state corresponds to the first excited state 1^1^A′ within the C_s_ symmetry group, in agreement with similar oscillator strengths for electron transitions and increased energy difference between the electronic states X^1^A′ and 1^1^A′ in this group compared to the energy difference between the states X^1^A_1_ and 1^1^B_1_ within the C_2v_ group (see excitation energies in Table 2). An analogous PJT interaction between X^1^A_1_ and 2^1^B_1_ within the C_2v_ group is also possible, but should be much weaker due to their higher energy difference. The JT stabilization energies (*E*_JT_) of the **IIa** and **IIb** models (Table 3) are the same, although the **IIa** model is slightly more stable (Table 1). These energies are comparable to the values published for Si_3_, CuF_3_, and CuO_6_ [34] or C_60_^3–^ [35]. Moreover, our results confirm that the enol–keto forms are more stable than the keto–keto forms, which is in agreement with the literature data [36].

The ground-state DFT energies of the C_1_ symmetry structures of the **IIa** and **IIb** model systems are higher than those of the C_s_ ones. Their existence can be explained by the PJT interaction between the excited electron states of the C_s_ symmetry structures (hidden JT effect [14]). The geometry optimization of both model systems of C_s_ symmetry in their second excited state 1^1^A″ leads to the structures with a single imaginary vibration of a″ symmetry (Appendix A). As C_1_ is its kernel subgroup K(C_s_, a″), it can be explained by the PJT interaction between 1^1^A″ and near A′ excited electron states within the C_s_ symmetry group. This is most probable with the 1^1^A′ state because their energy order is reversed compared to C_s_/X^1^A′ optimized structures, and their energy difference in the C_1_ symmetry structure increased most of all. The JT stabilization energy for the **IIa** model is lower, despite its higher stability compared to the **IIb** one. We have found curcumin enolate structures **IIIa** and **IIIb** of C_2v_ symmetry only without any imaginary frequency. Therefore, the pseudo-JT effect in these systems is too weak (lower vibronic coupling constant *F* and/or higher primary force constant *K* values) to be observed, despite the much lower excitation energies than in the keto–keto and enol–keto forms of curcumin (a much lower Δ value is demanded according to Equation (3)).

The experimental absorption spectrum of the curcumin enol–keto forms recorded in acetonitrile and methanol exhibits a broad band with maximum absorbance peaks at 418 nm (2.97 eV) and 422 nm (2.94 eV) [12], respectively, which could be assigned to low π→π* excitations [37]. The values of the calculated gas-phase TD-M062X excitation energies corresponding to the S_0_→S_1_ transitions for **IIa** and **IIb** (see Table 2) are overestimated by approximately 0.5 eV. The TD-B3LYP(SMD = water)/aug-cc-pVDZ//M062X/cc-pVDZ electron transitions for the studied forms (see Appendix A) exhibit lower excitation energies related to the TD-M062X ones. For example, the calculated energies of 2.73 eV (454 nm) and 2.70 eV (459 nm) for the C_s_ symmetry are in better agreement with the available experiment for the enol–keto forms (see Figure 2). The experimental spectrum for keto–keto forms **Ia** and **Ib** measured in aqueous solution [38] also shows a single peak at 340 nm (3.65 eV). In this case, minimal differences between experiment and theory were found for the third and fourth TD-B3LYP electron transitions with high oscillator strengths, i.e., 373 nm (3.33 eV) and 417 nm (2.97 eV), respectively. In organic solvents under strong basic conditions, the deprotonated sample absorbs at a longer wavelength with a broad peak at ≈ 480 nm [39]. The predicted corresponding excitation energies are 410 nm (3.02 eV) and 412 nm (3.01 eV) for **IIIa** and **IIIb**, respectively. 

Useful information on the electronic structure of enol–keto and keto–keto tautomers could be deduced from the analysis of molecular orbitals that contribute to the lowest electronic transition with significant oscillator strength. The calculated frontier orbital distributions for the selected structures, computed at the B3LYP(SMD = water)/aug-cc-pVDZ//M062X/cc-pVDZ level with an iso-value of 0.05 atomic units, are presented in Figure 3, Figure 4 and Figure 5. In the case of keto–keto forms, the electron clouds are either delocalized only over one-half of the molecular length (see **Ia**) or well separated (see **Ib**). For enol–keto forms, the electrons are excited from the enolic part to the center. Interestingly, enolic **IIIa** and **IIIb** forms have significant differences in the shapes of the HOMO orbitals. All the depicted transitions have a π→π* character.

## 4. Methods

Gaussian16 software was used for all quantum-chemical calculations [40]. Geometries of various forms of curcumin in singlet ground or excited states were optimized within C_2v_, C_2_, C_s_, or C_1_ symmetries using the M062X hybrid functional [41] with standard cc-pVDZ basis sets for all atoms from the Gaussian library. The stability of the obtained geometry was checked by vibrational analysis in the absence of imaginary vibrations. The Time-dependent DFT (TD-DFT) treatment [42,43] for up to 30 vertical electronic states was used for excited state calculations. For comparison, optical transitions were also calculated using the 3-parameter hybrid B3LYP functional [44] with standard aug-cc-pVDZ basis sets for all atoms of the Gaussian library in aqueous solutions within an SMD solvent model approximation (denoted as TD-B3LYP(SMD = water)/aug-cc-pVDZ//M062X/cc-pVDZ) [45]. MOLDRAW software [46] was used for geometry manipulation and visualization purposes. Molecular orbitals were drawn using Molekel software [47].

## 5. Conclusions

The systematic theoretical study of symmetric curcumin tautomers was performed using the hybrid functional granting reliable results on the structure of organic compounds in our previous studies. According to Jacquemin and Adamo [48], this functional is a reasonable choice for computation of long oligomers with alternating bond lengths. The symmetry point groups were identified for the investigated keto–keto, enol–keto, and enolate forms. Based on the normal mode analysis, unstable geometries with corresponding symmetries were determined. The indicated symmetry descent was ascribed to the PJT vibronic interaction between the ground electron state and the selected excited states. 

It can be expected that the use of other DFT functionals—with or without dispersion correction—should not cause any qualitative differences in our conclusions. The discrepancy between M062X and B3LYP in describing electronic excitation with a small amount of charge transfer character is well known. The computed wavelengths are systematically guided by the percentage of exact exchange included at intermediate interelectronic distances [49].

However, solvent-effect inclusion improves the agreement of TD-DFT results with experimental electron spectra, but forbids our treatment that uses imaginary vibrations of PJT unstable structures to predict the symmetry of PJT active electronic states. Polarizable continuum models create the solute cavity via a set of overlapping spheres that are approximated by polyhedron with non-vanishing edges. Due to inconsistency of the symmetry of such cavities with the molecular symmetry (especially for large molecules), additional imaginary vibrations are produced if the molecular symmetry is to be preserved during geometry optimization within integrals and their derivatives evaluation (keyword Symmetry) [40].

The photobiological action of curcumin can be elucidated by the deactivation mechanisms of its first excited singlet state with intramolecular proton transfer between the enol and keto moiety as the leading nonradiative deactivation pathway [50]. Our theoretical results demonstrated the potential significance of the PJT effect, which may modulate the setting of electronic and vibration (vibronic) energy levels upon photodynamic processes. We have shown that this effect leads to stable structures of the different symmetries of enol and keto forms. This might significantly affect the deactivation rate. Although our study does not directly address photodynamics and related photoprotection problems, the results obtained may rationalize the photoprotection action and activity of naturally occurring dyes. Our findings open new ways of studying and understanding the photodynamic processes of (quasi)symmetric organic molecules in plants and fungi. Further experimental and theoretical studies in this field are needed.

## Figures and Tables

**Figure 1 molecules-28-02946-f001:**
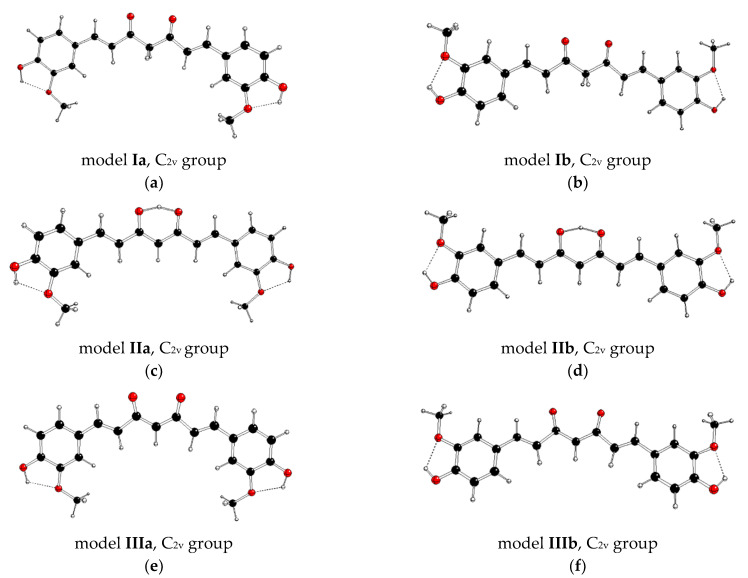
M062X/cc-pVDZ optimized curcumin structures of C_2v_ group of the neutral diketo forms (**a**,**b**), neutral enol forms (**c**,**d**), and anionic enolate forms (**e**,**f**) (C—black, O—red, H—white).

**Figure 2 molecules-28-02946-f002:**
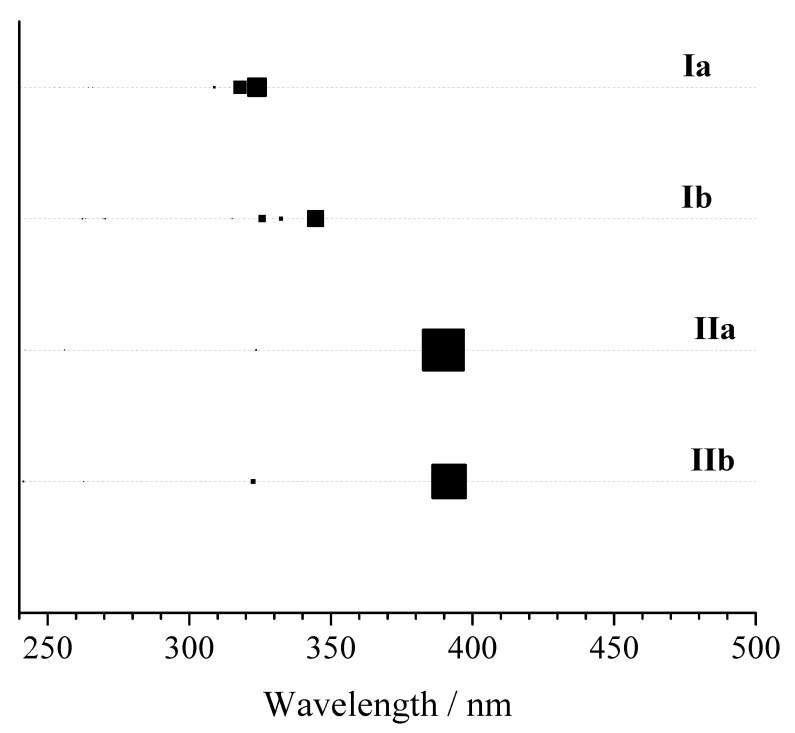
TD-B3LYP(SMD = water)/aug-cc-pVDZ//M062X/cc-pVDZ absorption wavelengths and oscillator strengths predicted for models **Ia** (C_2_ symmetry), **Ib** (C_2_ symmetry), **IIa** (C_s_ symmetry), and **IIb** (C_s_ symmetry). The size of the square symbols reflects the oscillator strength.

**Figure 3 molecules-28-02946-f003:**
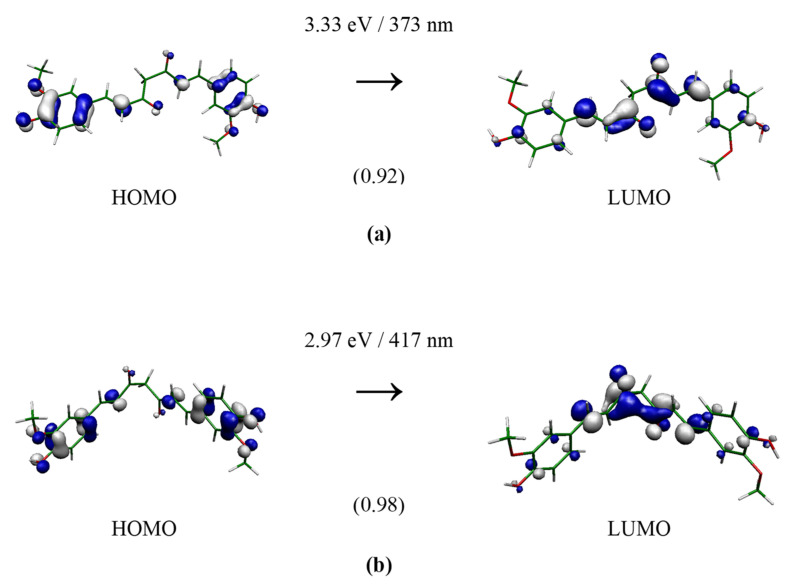
TD-B3LYP(SMD = water)/aug-cc-pVDZ//M062X/cc-pVDZ electron transition forming 2^1^B state of curcumin keto-form **Ia** (**a**) and **Ib** (**b**) of C_2_ group. The depicted orbital isosurface is 0.05 a.u. The oscillator strengths (in atomic units) are in parentheses.

**Figure 4 molecules-28-02946-f004:**
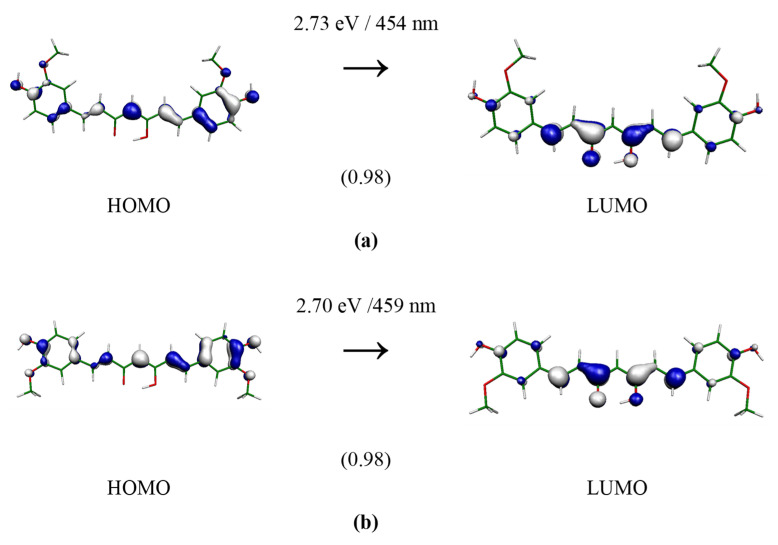
TD-B3LYP(SMD = water)/aug-cc-pVDZ//M062X/cc-pVDZ electron transition forming 1^1^A’ state of curcumin keto-form **Iia** (**a**) and **Iib** (**b**) of C_s_ group. The depicted orbital isosurface is 0.05 a.u. The oscillator strengths (in atomic units) are in parentheses.

**Figure 5 molecules-28-02946-f005:**
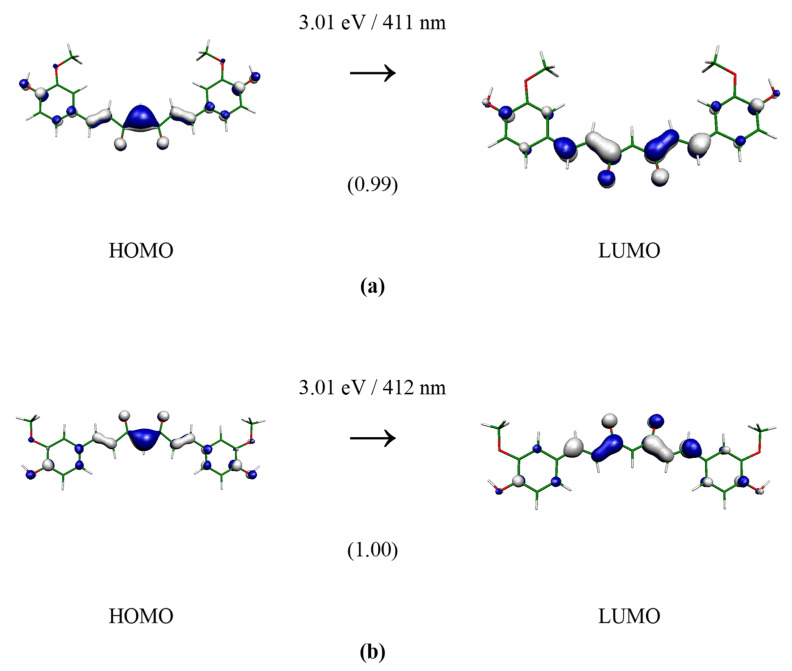
TD-B3LYP(SMD = water)/aug-cc-pVDZ//M062X/cc-pVDZ electron transition forming 1^1^B_1_ state of curcumin keto-form **IIIa** (**a**) and **IIIb** (**b**) of C_2v_ group. The depicted orbital isosurface is 0.05 a.u. The oscillator strengths (in atomic units) are in parentheses.

**Table 1 molecules-28-02946-t001:** Selected structural data and relative DFT energies (in kJ/mol) of optimized molecules for different electronic states. Stable structures are indicated by bold text. The dihedral angles (in degrees) are denoted by Greek symbols, distances *d* are in Å.

Model	Form	C=O vs. OCH_3_ Conformation	Group	State	DFT Energy	Imaginary Vibration	Θ(OC-CO) (Central)	Ω (H_3_CO-OCH_3_)	*d*(O-H) (Central)
**Ia**	keto–keto	*anti*, *anti*	C_2v_	X^1^A_1_	0.00	a_2_	0.0	0.0	-
			**C_2_**	**X^1^A**	−19.88	-	–111.1	–155.0	-
**Ib**	keto–keto	*syn*, *syn*	C_2v_	X^1^A_1_	−0.24	a_2_	0.0	0.0	-
			**C_2_**	**X^1^A**	−28.15	-	167.7	179.4	-
**IIa**	enol–keto	*anti*, *anti*	C_2v_	X^1^A_1_	−38.12	b_1_	0.0	0.0	1.289 (2×)
			**C_s_**	**X^1^A′**	−59.73	-	0.0	0.0	1.023 1.522
			C_s_	1^1^A″	−26.81	a″	0.0	0.0	0.969 1.913
			**C_1_**	**X^1^A**	−45.50	-	0.7	38.9	1.029 1.499
**IIb**	enol–keto	*syn*, *syn*	C_2v_	X^1^A_1_	−36.78	b_1_	0.0	0.0	1.289 (2×)
			**C_s_**	**X^1^A′**	−58.42	-	0.0	0.0	1.023 1.521
			C_s_	1^1^A″	−25.47	a″	0.0	0.0	0.969 1.915
			**C_1_**	**X^1^A**	−48.18	-	0.3	6.5	1.030 1.493
**IIIa**	enolate	*anti*, *anti*	**C_2v_**	**X^1^A_1_**	1438.35	-	0.0	0.0	-
**IIIb**	enolate	*syn*, *syn*	**C_2v_**	**X^1^A_1_**	1441.61	-	0.0	0.0	-

**Table 2 molecules-28-02946-t002:** Symmetry group/electronic ground state (GS), excitation energies (*E*_exc_) in eV and oscillator strengths (*f*) in atomic units for the corresponding electron transitions of the excited states for the optimized structures in relevant electron states (PJT active electron states in bold).

Model	Group/GS	Excited el. State	E_exc_	*f*	Group/GS	Excited el. State	E_exc_	*f*
**Ia**	C_2v_/X^1^A_1_	1^1^B_1_	3.443	0.00	C_2_/X^1^A	1^1^B	3.616	0.00
		**1^1^A_2_**	**3.621**	0.00		1^1^A	3.688	0.01
		1^1^B_2_	4.191	1.38		2^1^B	4.154	1.39
		1^1^A_1_	4.378	0.13		2^1^A	4.340	0.10
		2^1^B_2_	4.900	0.08		3^1^B	4.714	0.00
**Ib**	C_2v_/X^1^A_1_	1^1^B_1_	3.441	0.00	C_2_/X^1^A	1^1^B	3.484	0.02
		**1^1^A_2_**	**3.621**	0.00		1^1^A	3.699	0.00
		1^1^B_2_	4.171	1.34		2^1^B	3.960	0.91
		1^1^A_1_	4.353	0.13		2^1^A	4.190	0.32
		2^1^B_2_	4.867	0.02		3^1^B	4.813	0.01
**IIa**	C_2v_/X^1^A_1_	**1^1^B_1_**	**3.388**	1.75	C_s_(σ_v_)/X^1^A′	1^1^A′	3.475	1.81
		1^1^B_2_	4.048	0.00		**1^1^A″**	**3.850**	0.00
		1^1^A_1_	4.164	0.07		2^1^A′	4.159	0.08
		**2^1^B_1_**	**4.472**	0.07		3^1^A′	4.610	0.02
		2^1^A_1_	4.770	0.00		4^1^A′	4.792	0.00
**IIa**	C_s_/1^1^A″	1^1^A′	3.371	1.77	C_1_/X^1^A	1^1^A	3.446	1.40
		**1^1^A″**	**3.217**	0.00		2^1^A	3.884	0.00
		2^1^A′	3.957	0.13		3^1^A	4.177	0.19
		3^1^A′	4.551	0.02		4^1^A	4.583	0.05
		4^1^A′	4.687	0.01		5^1^A	4.797	0.00
**IIb**	C_2v_/X^1^A_1_	**1^1^B_1_**	**3.372**	1.78	C_s_(σ_v_)/X^1^A′	1^1^A′	3.459	1.83
		1^1^B_2_	4.044	0.00		**1^1^A″**	**3.848**	0.00
		1^1^A_1_	4.120	0.07		2^1^A′	4.118	0.07
		**2^1^B_1_**	**4.441**	0.04		3^1^A′	4.593	0.01
		2^1^A_1_	4.790	0.00		4^1^A′	4.800	0.01
**IIb**	C_s_/1^1^A″	1^1^A′	3.352	1.80	C_1_/X^1^A	1^1^A	3.424	1.48
		**1^1^A″**	**3.216**	0.00		2^1^A	3.892	0.00
		2^1^A′	3.916	0.11		3^1^A	4.149	0.17
		3^1^A′	4.536	0.02		4^1^A	4.573	0.05
		4^1^A′	4.695	0.02		5^1^A	4.810	0.00
**IIIa**	C_2v_/X^1^A_1_	1^1^B_2_	2.952	0.00				
		1^1^B_1_	3.045	0.72				
		1^1^A_2_	3.612	0.00				
		1^1^A_1_	3.637	0.01				
		2^1^B_1_	4.404	0.00				
**IIIb**	C_2v_/X^1^A_1_	1^1^B_2_	2.955	0.00				
		1^1^B_1_	3.056	0.79				
		1^1^A_2_	3.621	0.00				
		1^1^A_1_	3.680	0.00				
		2^1^B_1_	4.373	0.02				

**Table 3 molecules-28-02946-t003:** PJT stabilization energies E_JT_ for the relevant electron states of the stable optimized structures of JT subgroups related to the ones of their parent groups and electron states (the preserved σ_v_ mirror plane is coincident with the curcumin molecular plane).

Model	Parent Group		Subgroup		E_JT_
	Symmetry	Electron State	Symmetry	Electron State	/eV
**Ia**	C_2v_	X^1^A_1_	C_2_	X^1^A	0.206
**Ib**	C_2v_	X^1^A_1_	C_2_	X^1^A	0.289
**IIa**	C_2v_	X^1^A_1_	C_s_(σ_v_)	X^1^A′	0.224
	C_s_	1^1^A″	C_1_	X^1^A	0.194
**IIb**	C_2v_	X^1^A_1_	C_s_(σ_v_)	X^1^A′	0.224
	C_s_	1^1^A″	C_1_	X^1^A	0.235

## Data Availability

Data is contained within the article or supplementary material.

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
