# Peer review of "How Does Pseudo-Jahn-Teller Effect Induce the Photoprotective Potential of Curcumin?"

_molecules, 2023, doi:10.3390/molecules28072946_

Round 1

Reviewer 1 Report

In this work, the authors performed DFT calculations to systematically study the Pseudo Jahn-Teller effect in curcumine. The PJT effect in curcumine is revealed by the symmetry analysis of the ground state and excited states of different possible structures. This work is helpful to understand the energy level and the symmetry change in the photodynamic process of curcumin, and can be interesting to both computational and experimental chemistry communities. In addition, I appreciate that the author uses the reason functionals and basis set in the DFT calculations. The Dunning basis set is a more proper choice than the outdated Pople basis set. Thus, I recommend its publication after minor revisions and Molecules is certainly well targeted.

Here are the comments/questions I would like to ask the authors to address before the publication.

1. Page 3, line 87

The term “adiabatic potential surfaces (APS)” is already defined in previous section.

2. Page 4, line 127

The term “pseudo-Jahn-Teller effect (PJTE)” is already defined in previous section.

3. Page 6, line 201

“On the other hand, an approximately four times larger EJT was predicted for C4H4 is 0.78 eV”

The reference for this result is missing.

4. Page 6, Table 1

It is interesting that IIIa and IIIb have no imaginary frequencies, despite they have the same symmetry as other systems in the table. Can authors explain the reason these two structures with higher energies having no imaginary frequency?

5. Page 7, Table 3

In this table, after reducing the symmetry, the PJT stabilization energy of the system IIa is decreased, which are different from the other two systems. Can authors provide an explanation to this?

6. Page 9, line 237

The term “TD-B3LYP/M062X” is confusing. Does this indicate the TD-B3LYP calculation is performed on top of the M062X ground state calculation, or only the structure is optimized with M062X but the ground state calculation is by B3LYP?

7. Page 9, line 258

The basis set used for the TDDFT calculations is not mentioned in the Method section. Did authors use the cc-pVDZ basis set? If yes, the basis set for TDDFT is not well converged. The augmented basis set is necessary for TDDFT calculations.

Author Response

How Does Pseudo-Jahn-Teller Effect Induce The Photoprotective Potential of Curcumin?

Dagmar Štellerová, Vladimír Lukeš and Martin Breza

Replies to Reviewer 1

In this work, the authors performed DFT calculations to systematically study the Pseudo Jahn-Teller effect in curcumine. The PJT effect in curcumine is revealed by the symmetry analysis of the ground state and excited states of different possible structures. This work is helpful to understand the energy level and the symmetry change in the photodynamic process of curcumin, and can be interesting to both computational and experimental chemistry communities. In addition, I appreciate that the author uses the reason functionals and basis set in the DFT calculations. The Dunning basis set is a more proper choice than the outdated Pople basis set. Thus, I recommend its publication after minor revisions and Molecules is certainly well targeted.

Here are the comments/questions I would like to ask the authors to address before the publication.

Comment: 1. Page 3, line 87

The term “adiabatic potential surfaces (APS)” is already defined in previous section.

Reply: Amended.

Comment: 2. Page 4, line 127

The term “pseudo-Jahn-Teller effect (PJTE)” is already defined in previous section.

Reply: Amended.

Comment: 3. Page 6, line 201

“On the other hand, an approximately four times larger EJT was predicted for C4H4 is 0.78 eV”

The reference for this result is missing.

Reply: The sentence has been deleted. We apologize for this error.

Comment: 4. Page 6, Table 1

It is interesting that IIIa and IIIb have no imaginary frequencies, despite they have the same symmetry as other systems in the table. Can authors explain the reason these two structures with higher energies having no imaginary frequency?

Reply: The stability of these deprotonated anionic model systems of C2v symmetry could be explained by lower   vibronic coupling constant F and/or higher primary force constant K values, which implies a much lower Δ value as required according to Eq. (3)

Comment: 5. Page 7, Table 3

In this table, after reducing the symmetry, the PJT stabilization energy of the system IIa is decreased, which are different from the other two systems. Can authors provide an explanation to this?

Reply: We have no explanation.

Comment: 6. Page 9, line 237

The term “TD-B3LYP/M062X” is confusing. Does this indicate the TD-B3LYP calculation is performed on top of the M062X ground state calculation, or only the structure is optimized with M062X but the ground state calculation is by B3LYP?

Reply: The term has been explained in Methods.

Comment: 7. Page 9, line 258

The basis set used for the TDDFT calculations is not mentioned in the Method section. Did authors use the cc-pVDZ basis set? If yes, the basis set for TDDFT is not well converged. The augmented basis set is necessary for TDDFT calculations.

Reply: All TD-B3LYP data have been recalculated using aug-cc-pVDZ basis sets and the SMD solvent model for aqueous solutions as mentioned in the Method section.

We thank the reviewer for valuable comments.

Reviewer 2 Report

This paper explores the molecular and electronic structure of curcumin, a natural dye present in turmeric. The authors investigated high-symmetry gas-phase tautomers and their deprotonated forms in various symmetry groups. Additionally, they used the Pseudo Jahn Teller effect to explain why lower-symmetry structures are more stable. 

The manuscript shows several drawbacks and I’m afraid, but I cannot recommend it for publication in the present form.

1) The authors compared excitation energies computed in gas phase with experimental results obtained in solvent (i.e, water). This comparison is not fair. The authors should compare the gas phase results with higher level calculations in gas phase, or compute the excitation energies in solvent. Indeed, curcumin have different carbonyl and hydroxyl groups that can do explicit hydrogen bond with water molecules influencing the excitation energies. At least a comment on the solvent effects should be included in the paper to explain the discrepancy between experimental and computed values. Several works have shown the importance of including solvent effects to describe the properties of similar dyes both in terms of excitation energy  (J. Comput. Chem., 2020, 41, 2228International Journal of Quantum Chemistry 119 (1), e25719Journal of chemical theory and computation 13 (5), 2159-2171) and excited state reactivity (J. Chem. Theory and Comput., 2020, 16, 7033; The Journal of Physical Chemistry B 125 (45), 12539-12551). These references should be included. 

2) Why did the authors use M06-2X for the ground state geometries optimization? This kind of functional with 50% of HF exchange could lead to an incorrect description of the bond length alternation of the molecular skeleton. A justification should be provided.

3) The discrepancy between M06-2X and B3LYP in describing electronic excitation with small amount of charge transfer character is well known in literature (i.e. Journal of chemical theory and computation 4 (1), 123-135). A comment could be added based on this reference.

4) It’s extremely hard to relate the photoprotection action to the Pseudo Jan Teller effect. The authors should explain more in detail this connection, which could be the main result to justify the publication on Molecules. Several experimental and theoretical works have tried to unveil a correlation between vibrational fingerprints and electronic properties; for example, what vibrational mode can lead to a conical intersection ( J. Phys. Chem. A, 2018, 122, 14, 3594; Chem. Sci., 2021, 12, 8058; Phys. Chem. Chem. Phys., 2019, 21, 3606; Phys. Chem. Chem. Phys., 2015, 17, 9231–9240). Is this the connection the authors are trying to make?

Minor points:

1) Overall, the readability of the paper can be improved.

2)   Why absolute Hartree energy are reported in table 1? Would not be better to report relative energies of the conformers in kcal/mol?

 3) A picture showing molecular orbitals involved in the bright excitation could be moved in the main paper (at least for one conformer). 

Author Response

How Does Pseudo-Jahn-Teller Effect Induce The Photoprotective Potential of Curcumin?

Dagmar Štellerová, Vladimír Lukeš and Martin Breza

Replies to Reviewer 2

This paper explores the molecular and electronic structure of curcumin, a natural dye present in turmeric. The authors investigated high-symmetry gas-phase tautomers and their deprotonated forms in various symmetry groups. Additionally, they used the Pseudo Jahn Teller effect to explain why lower-symmetry structures are more stable. 

 The manuscript shows several drawbacks and I’m afraid, but I cannot recommend it for publication in the present form.

 Comment: 1) The authors compared excitation energies computed in gas phase with experimental results obtained in solvent (i.e, water). This comparison is not fair. The authors should compare the gas phase results with higher level calculations in gas phase, or compute the excitation energies in solvent. Indeed, curcumin have different carbonyl and hydroxyl groups that can do explicit hydrogen bond with water molecules influencing the excitation energies. At least a comment on the solvent effects should be included in the paper to explain the discrepancy between experimental and computed values. Several works have shown the importance of including solvent effects to describe the properties of similar dyes both in terms of excitation energy  (J. Comput. Chem., 2020, 41, 2228; International Journal of Quantum Chemistry 119 (1), e25719; Journal of chemical theory and computation 13 (5), 2159-2171) and excited state reactivity (J. Chem. Theory and Comput., 2020, 16, 7033; The Journal of Physical Chemistry B 125 (45), 12539-12551). These references should be included. 

Reply: All TD-B3LYP data have been recalculated using aug-cc-pVDZ basis sets and the SMD solvent model for aqueous solutions, as mentioned in the Methods section.

Comment: 2) Why did the authors use M06-2X for the ground state geometries optimization? This kind of functional with 50% of HF exchange could lead to an incorrect description of the bond length alternation of the molecular skeleton. A justification should be provided.

Reply: In J. Chem. Theory Comput. 2011, 7, 369–376, the M06-2X functional is recommended for a correct description of bond length alternation in long oligomers.

Comment: 3) The discrepancy between M06-2X and B3LYP in describing electronic excitation with small amount of charge transfer character is well known in literature (i.e. Journal of chemical theory and computation 4 (1), 123-135). A comment could be added based on this reference.

Reply: Amended 

Comment: 4) It’s extremely hard to relate the photoprotection action to the Pseudo Jan Teller effect. The authors should explain more in detail this connection, which could be the main result to justify the publication on Molecules. Several experimental and theoretical works have tried to unveil a correlation between vibrational fingerprints and electronic properties; for example, what vibrational mode can lead to a conical intersection ( J. Phys. Chem. A, 2018, 122, 14, 3594; Chem. Sci., 2021, 12, 8058; Phys. Chem. Chem. Phys., 2019, 21, 3606; Phys. Chem. Chem. Phys., 2015, 17, 9231–9240). Is this the connection the authors are trying to make?

Reply: The photobiological action of curcumin can be elucidated by the deactivation mechanisms of its first excited singlet state with intramolecular proton transfer between the enol and keto moiety as the leading nonradiative deactivation pathway. Its rate should be significantly influenced by the PJT effect (see Conclusions).

Minor points:

Comment: 1) Overall, the readability of the paper can be improved.

Reply: We have inserted some explanations into Theory.

Comment: 2)   Why absolute Hartree energy are reported in table 1? Would not be better to report relative energies of the conformers in kcal/mol?

Reply: The absolute energies have been recalculated to relative energies in kJ/mol units.

 Comment: 3) A picture showing molecular orbitals involved in the bright excitation could be moved in the main paper (at least for one conformer). 

 Reply: Amended.

Round 2

Reviewer 2 Report

The paper can be accepted.